# Comparison and Characterization of a Cell Wall Invertase Promoter from Cu-Tolerant and Non-Tolerant Populations of *Elsholtzia haichowensis*

**DOI:** 10.3390/ijms22105299

**Published:** 2021-05-18

**Authors:** Rongxiang Liu, Jing Zhao, Zhongrui Xu, Zhiting Xiong

**Affiliations:** 1Hubei Biomass-Resource Chemistry and Environmental Biotechnology Key Laboratory, School of Resource and Environmental Science, Wuhan University, Wuhan 430079, China; liurongxiang@whu.edu.cn (R.L.); zhaojing@whu.edu.cn (J.Z.); xuzhongrui@njau.edu.cn (Z.X.); 2Qiandongnan Vocational and Technical College for Nationalities, Kaili 556000, China

**Keywords:** cell wall invertase, *Elsholtzia haichowensis*, promoter, populations, salicylic acid, copper, sugar, phytohormone, *cis* elements

## Abstract

Cell wall invertase (CWIN) activity and the expression of the corresponding gene were previously observed to be significantly elevated in a Cu-tolerant population of *Elsholtzia haichowensis* relative to a non-tolerant population under copper stress. To understand the differences in *CWIN* gene regulation between the two populations, their *CWIN* promoter *β*-glucuronidase (GUS) reporter vectors were constructed. GUS activity was measured in transgenic Arabidopsis in response to copper, sugar, and phytohormone treatments. Under the copper treatment, only the activity of the *CWIN* promoter from the Cu-tolerant population was slightly increased. Glucose and fructose significantly induced the activity of *CWIN* promoters from both populations. Among the phytohormone treatments, only salicylic acid induced significantly higher (*p* < 0.05) activity of the Cu-tolerant *CWIN* promoter relative to the non-tolerant promoters. Analysis of 5′-deletion constructs revealed that a 270-bp promoter fragment was required for SA induction of the promoter from the Cu-tolerant population. Comparison of this region in the two *CWIN* promoters revealed that it had 10 mutation sites and contained CAAT-box and W-box *cis*-elements in the Cu-tolerant promoter only. This work provides insights into the regulatory role of SA in *CWIN* gene expression and offers an explanation for differences in *CWIN* expression between *E. haichowensis* populations.

## 1. Introduction

Sucrose (Suc) is the major product of photosynthesis in the source tissues of higher plants; it is transported to sink tissues to sustain cell metabolism, growth, and environmental stress tolerance [1,2]. Two enzymes catalyze sucrose cleavage in higher plants: sucrose synthase (Sus, EC 2.4.1.13) and invertase (IN, EC 3.2.1.26) [3]. Sucrose synthase converts sucrose into UDP-glucose and fructose, whereas invertase cleaves sucrose into glucose and fructose [1]. Based on their pH optima, solubility, and sub-cellular locations, invertases are classified into vacuolar invertases (VINs), cell wall invertases (CWINs), and cytoplasmic invertases (CINs) [1,3]. CINs are located in the cytoplasm, where they maintain cytosolic sugar homeostasis for cellular function [3,4]. VINs are located in the vacuole, where they function primarily to control cell enlargement, osmoregulation, sugar composition of fruits and storage organs, and the response to cold and other stimuli [1,5]. CWINs are found in the apoplast and ionically bound to the cell wall. They can irreversibly cleave sucrose when it is unloaded from the sieve elements of the phloem into the apoplast, and the resulting hexoses are transported into sink cells for use [4]. Since the substrates and products of these enzymes are both nutrient and signaling molecules, CWIN participate in many aspects of plant physiology, controlling growth, differentiation, development, reproduction, and adaptation to biotic and abiotic stresses [3,6,7]. Recently, a series of studies have found that heavier seed weight in maize, rice, and Arabidopsis [8], greater fruit set in tomato [9,10], heat tolerance in tomato [11], cold tolerance in rice [12], and drought tolerance in wheat [13] are attributable to increased CWIN activity. Therefore, studying *CWIN* gene regulation is helpful for understanding the molecular mechanisms of plant high yield or environmental tolerance. *Elsholtzia haichowensis* (or *E. splendens*) is a well-known valuable indicator of copper (Cu) mines [14,15]. Two *E. haichowensis* populations of differing Cu tolerance are found in areas along the middle and lower streams of the Yangtze River in China. Cu-tolerant populations are found at the famous old Cu mine at Tonglushan Hill in Hubei Province, and non-tolerant populations are found in non-metalliferous soils of Hongan county in Hubei Province [5,16]. In our previous studies, we noted an interesting phenomenon: the Cu-tolerant populations allocated more biomass to the roots than non-tolerant populations under Cu stress [5,17]. In addition, root CWIN activity and *CWIN* gene expression were significantly higher (*p* < 0.05) in the Cu-tolerant population than in the non-tolerant population under Cu stress [17,18]. Similarly, our previous studies demonstrated higher CWIN activity in Cu-tolerant populations of *Rumex japonicus* [19], *R. dentatus* [20,21], and *E. haichowensis* [5,18] relative to non-tolerant populations. However, the *CWIN* genes from the two populations had extremely similar deduced amino acid sequences and predicted 3D structures (data not shown). We do not know what causes the difference in *CWIN* expression between the two populations under Cu stress.

The *CWINs* have been shown to be regulated by a variety of internal and environmental stimuli, including sugars, phytohormones, wounding, heavy metals, and others [6,22,23]. It is well known that gene expression is regulated primarily at the transcriptional level and initiated by the binding of specific transcription factors (TFs) to *cis*-regulatory sites in gene promoters [24]. Promoters are key regulatory DNA elements located upstream of the gene coding region [25]. TFs recognize specific regulatory DNA elements and coordinate the timing and strength of gene expression during development, differentiation, growth, and stress conditions [26]. Forty seven percent of gene *cis*-regulatory elements differ between two *Panicum hallii* ecotypes of contrasting drought tolerance, leading to differences in gene expression in response to environmental stimuli [27]. Gould et al. also reported that 79% of gene promoters (*cis*-regulatory elements) differ between two locally adapted ecotypes of yellow monkeyflower [28]. These results suggest that interspecies differences in promoter *cis*-regulatory elements may play an important role in environmental adaptation. Therefore, we speculated that differences in *CWIN* expression between Cu-tolerant populations and non-tolerant populations might be related to differences in promoter *cis*-regulatory elements. To test our hypothesis, the full-length promoter and serial deletions from its 5′ end were cloned into the pCAMBIA1301 plant expression vector by replacing the *CaMV35S* promoter upstream of the *GUS* reporter gene. Transgenic Arabidopsis were obtained by the floral dip method, treated with Cu, sugar, and phytohormones, and their GUS activity and *GUS* gene expression were compared. The study had two aims: (1) to compare and characterize the expression of the *CWIN* promoters from the two populations in response to Cu, sugar, and hormone treatments, and (2) to identify specific promoter regions that may be differentially regulated between the populations and potential *cis* elements unique to the Cu-tolerant population. Our study contributes to understanding the molecular mechanisms that underlie differences in *CWIN* expression between Cu-tolerant and non-tolerant populations of *E. haichowensis*.

## 2. Results

### 2.1. Comparative Bioinformatics Analysis of Promoters

To analyze differences in *CWIN* expression between the two populations under Cu stress, we used the Clustal W multiple sequence aligner to compare the full-length sequences of the Cu-tolerant promoter (*EhCcwINVP*) and the non-tolerant promoter (*EhNcwINVP*) of *E. haichowensis*. Thirty-two differences were identified between the two promoters, including a number of single nucleotide polymorphisms (SNPs) and insertion/deletions (InDels) (Figure 1A). Based on functional annotation of the *cis* elements using the PlantCARE database, we classified the *cis* elements into stress-, sugar-, phytohormone-, and plant development-related elements. Among the stress-related cis-elements, Cu responsiveness (CuRE), sugar responsiveness (TATCCA, W-box, SP8b), light responsiveness (BOX-4, G-box), low-temperature responsiveness (LIR), drought inducibility (MBS), and pathogen or elicitor induction responsiveness (W-box) elements were detected in both *EhNcwINVP* and *EhCcwINVP*. These *cis* elements were consistent in number and relative position between the two promoters (Figure 1A). The phytohormone-related cis elements included SA responsiveness (TCA-element), GA responsiveness (CAREOSREP1, TATC-box), cytokinin responsiveness ABA responsiveness (ABRE), and auxin response factor binding site (ARFAT) elements. The ABRE and ARFAT elements had different relative positions in the promoters from the two populations. Plant development-related *cis* elements included the sugar-repressive element (SRE) and the circadian control (CIRCADIAN) element. The SRE *cis* element was found only in the *EhCcwINVP* promoter from the Cu-tolerant population. Taken together, our analyses revealed a high degree of sequence homology between *EhNcwINVP* and *EhCcwINVP* in some regions and variation in others. (CPBCSPOR), ethylene responsiveness (GCC-box).

### 2.2. Promoter–Reporter Constructs and GUS Expression Analysis in Transgenic Arabidopsis

To measure promoter activity, the full-length sequences of *EhCcwINVP* and *EhNcwINVP* were fused to the coding sequence of β-glucuronidase (GUS) to construct the *EhCcwINVP:GUS* (−1729 to ATG, C0) and *EhNcwINVP:GUS* (−1724 to ATG, N0) vectors. To further identify promoter fragments whose activity differed between C0 and N0, four 5′-deleted promoter fusions were constructed from C0: C1 (−1460 to ATG), C2 (−729 to ATG), C3 (−612 to ATG), and C4 (−209 to ATG) (Figure 1A,B). The strong, constitutive expression of the CaMV35S promoter (CaMV35S:GUS) was used as the positive control (35S), and the Arabidopsis wild type (no GUS gene) was used as the negative control (WT). GUS histochemical staining was performed on 35S, N0, C0, C1, C2, C3, and C4 transgenic Arabidopsis lines to characterize the expression patterns of the *CWIN* promoters. The results showed GUS activity in all organs of N0, C0, C1, C2, C3, and C4 lines, including seedlings, flowers, and silique pods (Figure 2). However, GUS staining was not detected in any tissues of the WT (negative control) (Figure 2, WT-a, WT-b, and WT-c). The 35S promoter (positive control) drove high levels of GUS activity throughout the whole plant (Figure 2, 35S-a, 35S-b, and 35S-c). These results demonstrated that the 35S, N0, C0, C1, C2, C3, and C4 promoter constructs could directly drive GUS expression in transgenic Arabidopsis.

### 2.3. Quantitative Analysis of GUS Activity Driven by Full-Length Promoters in Response to Cu, Sugar, and Hormone Treatments

Bioinformatics analysis showed that Cu, sugar, and hormone response elements were present in the full-length N0 and C0 promoters. Therefore, we speculated that these abiotic factors might regulate *CWIN* promoter expression.

For the Cu treatment, fourteen-day-old seedlings were transferred to solid 1/2 MS medium that contained 0.3 μM to 200 μM Cu^2+^, the GUS activity was measured after 24 h of treatment. In the C0 line, GUS activity was higher following exposure to 50–200 μM Cu relative to the control treatment (CK, 0.3 μM Cu^2+^); the differences between 100 and 150 μM Cu^2+^ and the CK were significant (*p* < 0.05). No significant differences were observed between the C0 and N0 lines at any Cu level, but the relative GUS activity of C0 was always higher than that of N0 and 35S (Figure 3A).

For the sugar treatment, transgenic Arabidopsis were treated with sucrose (Suc, 200 mM), glucose (Glc, 200 mM), fructose (Fru, 200 mM), fructose + glucose (F + G, 100 mM, 1:1), mannitol (Man 200 mM, osmotic control), and 1/2 MS liquid medium (no sugar, CK) GUS activity was generally higher in the C0, N0, and 35S plants that received sugar treatments than in their respective controls. As expected, GUS activity was significantly induced (*p* < 0.05) in both C0 and N0 lines after treatment with Fru, Fru + Glc, and Glc (Figure 3B), but the 35S lines showed no difference in GUS activity compared with their control (Figure 3B). There were also no significant differences (*p* < 0.05) in GUS activity between the C0 and N0 lines (Figure 3B). Thus, we concluded that CWIN regulation by glucose and fructose was shared by the two populations. In addition, there were no significant differences between the mannitol treatment (osmotic control) and the CK treatment (blank control) for C0 or N0, indicating that the *CWIN* promoter was not regulated by osmotic pressure.

For the hormone treatments, exogenous abscisic acid (ABA), auxin (IAA), gibberellin (GA), ethylene precursor (1-aminocyclopropane-1-carboxylic acid, ACC), methyl jasmonate (MeJA), salicylic acid (SA), and sterile water (control, CK) were applied to the transgenic Arabidopsis seedlings. GUS activity was slightly inhibited by ABA in the C0 and N0 lines but not in the 35S lines (Figure 3C). GUS activity was generally increased by SA, 6-BA, GA, IAA, and ACC treatment in the C0 and N0 lines relative to their respective controls (Figure 3C). We were particularly interested to note that SA treatment significantly increased the GUS activity of C0 relative to its control (*p* < 0.05) and also relative to that of the N0 and 35S promoter lines (*p* < 0.05) (Figure 3C). This result suggested that the *CWIN* promoter from Cu-tolerant populations was specifically induced by SA.

The effect of SA on plants depends on its concentration [29]. Therefore, we treated transgenic 35S, C0, and N0 Arabidopsis lines with a series of SA concentrations (0, 50, 100, 250, 500, and 1000 μM). In the C0 lines, GUS activity increased significantly (*p* < 0.05) at concentrations of SA from 100 to 1000 μM, but no such effect was observed in the N0 and 35S lines under the same SA concentrations. In fact, GUS activity of the 35S lines was significantly inhibited (*p* < 0.05) at 1000 μM SA (Figure 4). The greatest difference among treatments was observed at 100 μM SA, when GUS activity in the C0 lines averaged 1.45 times higher than that in the CK. These results further confirmed that the activity of the promoter from a Cu-tolerant *E. haichowensis* population can be induced by a range of SA concentrations.

### 2.4. Transcriptional Responses of Full-Length Promoters to SA Treatments

To further understand whether GUS activity reflected GUS gene expression in transgenic Arabidopsis under SA treatment, the C0, N0, and 35S seedlings were treated with 100 μM SA for 6, 12, and 24 h, and samples were collected and analyzed by qRT–PCR. *GUS* expression of C0 promoter was statistically significantly higher (*p* < 0.05) than N0, 35S, and their controls at 6 and 12 h (Figure 5). The highest expression in C0 (1.94 times than its control) was observed at 12 h (Figure 5). These results provide further evidence that SA can induce the activity of the *CWIN* promoter from a Cu-tolerant population.

### 2.5. Identification of the CWIN Promoter Region Affected by SA in EhCcwINVP

To identify specific regions in *EhCcwINV**P* that were potentially involved in the response to SA, we measured the GUS activity of four 5′ deletion constructs in response to 100 μM SA treatment. The GUS activities of the C1 to C4 constructs did not differ significantly (*p* < 0.05) from those of their respective untreated controls (Figure 6), and their relative GUS activities were similar to those of N0 and 35S under SA stress (Figure 3B). This result indicated that the key promoter region for SA regulation was present in C0 but absent in C1 to C4. C0 contained a 270-bp region (−1729 to −1460) that was deleted from C1–C4. Thus, our analysis suggested that these 270 bp constitute the key promoter region that is regulated by SA in the Cu-tolerant population. Since N0 was not regulated by SA (Figure 3B), we compared the 270-bp regions in *EhCcwINVP* and *EhNcwINVP* (Figure 1). The sequences showed 8% divergence, including including eight SNP sites and two InDel sites (Figure 1A). Moreover, two putative *cis* elements, a W-box (TTGAC) and a CAAT-box (CCAAT), were found only in the promoter from the Cu-tolerant population.

## 3. Discussion

Cell wall invertase (CWIN) hydrolyzes sucrose into fructose and glucose. It is involved in establishing the sink strength of various sink tissues to activate a cascade of defense responses and to mediate physiological adaptations [6,7]. Previous studies have reported that *CWIN* genes are regulated by heavy metals [30,31,32], sugars [23,33,34], and plant hormones [4,35]. The regulation of plant gene expression in response to environmental stress is complex and involves multiple regulatory layers, including transcriptional, post-transcriptional, post-translational, and metabolic regulation [36]. Promoters have crucial roles in the transcriptional regulation of gene expression [37]. In this work, we bioinformatically characterized the *CWIN* promoters from Cu-tolerant and non-tolerant populations of *E. haichowensis* (Figure 1A). We cloned the full-length promoters and several promoter fragments, constructed promoter–GUS reporter vectors, and used them to transform Arabidopsis (Figure 1B). GUS activity was measured in transgenic Arabidopsis in response to copper, sugar, and plant hormone treatments. The most striking difference between the promoters from Cu-tolerant and non-tolerant populations was their response to salicylic acid, and the promoter fragment that responds to salicylic acid in the Cu-tolerant promoter was identified.

### 3.1. Copper Does Not Induce Differential Expression of CWIN Promoters from the Cu-Tolerant and Non-Tolerant Populations

A copper response element (CuRE) has been identified in *Chlamydomonas reinhardtii* and has the core sequence GTAC [38]. This CuRE element was found in both *CWIN* promoters from the Cu-tolerant and non-tolerant populations (Figure 1A). Under copper stress, the activity of the promoter from the non-tolerant population (*EhNcwINVP*) showed little difference relative to the control (CK) (Figure 3A). A single copy of the core CuRE sequence appeared to be insufficient to drive increased promoter activity in response to copper, and similar observations have been confirmed in Arabidopsis [39] and the moss *Barbula unguiculata* [40]. Therefore, the activity of *EhNcwINVP* may not have been induced by Cu, because it contains only a single copy of CuRE. The activity of *EhCcwINVP* was higher than that of the control in the 25–200 μM Cu treatments, and this difference was significant (*p* < 0.05) for the 100 μM and 150 μM Cu treatments (Figure 3A). The enhancer effect may explain the increased activity of *EhCcwINVP* in response to Cu. An intron of the *Sh1* maize sucrose synthase gene was identified as an enhancer of gene expression; it markedly enhanced expression of the bacterial chloramphenicol transacetylase (*CAT)* marker gene in both monocot and dicot protoplasts when it was introduced into the 5′ region of a gene [41]. We speculate that there may be a copper-responsive enhancer on the promoter of *EhCcwINVP*. Another possible explanation for the increased activity of *EhCcwINVP* in response to copper may be promoter methylation modification. In our previous study, *EhNcwINVP* and *EhCcwINVP* contained a small number of methylation modifications sites that showed significant differences under Cu stress [42]. Villalobos et al. reported that *Pi* starvation in Arabidopsis induces significant changes in promoter methylation status, thereby modulating the expression of phosphate starvation-responsive (*PSR*) genes to promote root phosphorus [43]. Therefore, methylation modification would also affect the promoter expression. However, we did not observe a significant difference (*p* < 0.05) in promoter activity between *EhNcwINVP* and *EhCcwINVP* activity under copper stress (Figure 3A), which suggested that Cu did not induce differential expression of the *CWIN* promoters from the Cu-tolerant and non-tolerant populations.

### 3.2. Glucose and Fructose Significantly Induced the Activity of the CWIN Promoter in Both Cu-Tolerant and Non-Tolerant Populations

The sugar-regulated *CWIN* gene has been studied extensively. Roitsch et al. reported that CWIN activity and *CWIN* mRNA levels were induced by glucose, fructose, and sucrose in *Chenopodium rubrum* [33]. Similarly, the *CWIN* gene was up-regulated by glucose in Arabidopsis [44] and tobacco [34] and by sucrose in tomato [35]. Some studies have confirmed that CWIN hydrolyzes sucrose into hexose, which serves as an important signaling molecule that regulates *CWIN* expression [2,23]. In transcriptional regulation, a number of sugar response elements have been reported; these include sugar starvation elements such as the sugar-repressive element (SRE, GGATAA) [45], the GC box (GGAGAACCGGG), the G box (CTACGTG), and the TATCCA element [46,47]. Sugar induction promoter elements include SURE1 (AATAGAAAA), SURE2 (AATACTAAT) [48], SP8a (ACTGTGTA), SP8b (TACTATT) [49], the W-box [(T)TGAC(C/T)] [50], and the TGGACGG element [51]. Bioinformatics analysis showed that the TATCCA sugar starvation element was present in both *EhNcwINVP* and *EhCcwINVP*, but SRE was present only in *EhCcwINVP* (Figure 1A). The sugar induction promoter element SP8b was found in both promoters, and four W-boxes and six W-boxes were found in *EhCcwINVP* and *EhNcwINVP*, respectively (Figure 1A). In this study, we found that glucose (Glc), fructose (Fru), and glucose + fructose (Glc + Fru, 1:1) significantly induced the activity of both *EhNcwINVP* (N0) and *EhCcwINVP* (C0) (Figure 3B). The promoter activities of 35S and C0 increased slightly under sucrose treatment, but only that of N0 was significantly higher (*p* < 0.05) than its control (Figure 3B). In potato, nine *SRE* elements and three TATCCA elements were identified in the promoter of the vacuolar invertase gene *StvacINV1*, and its promoter activity decreased significantly in response to Suc and Glc treatments [52]. The sugar-repressive element SRE and a sugar-responsive W-box element (TGACT) were found in the promoter of the tea vacuolar invertase *CsINV5*, and its activity increased under Suc treatment [53]. The barley transcription factor SUSIBA2 has been reported to bind to SURE and W-box elements but not to the SP8a element in the isoamylase1 (*iso1*) promoter in response to Suc treatment [50]. Whether the significant induction of N0 by Suc is related to the absence of an SRE element in its promoter remains to be determined. Together, our results suggested that the W-box or SP8a elements but not the TATCCA element may act as core motifs to control the response to sugar. However, our data do not support sugar regulation as a direct factor responsible for the difference in *CWIN* promoter activity between the two populations.

### 3.3. SA Induces Differential Expression of CWIN Promoters from Two Populations

Our analysis suggested that several plant hormones could affect the expression of *CWIN*; IAA, 6-BA, ACC, and GA treatments caused a slight increase, whereas ABA treatment caused a slight decrease. The regulation of invertase expression by multiple phytohormones has been reviewed previously [4]. In particular, we noted that SA concentrations of 100–1000 μM significantly increased (*p* < 0.05) the activity of the *CWIN* promoter from the Cu-tolerant population (C0) but had no effect on the promoter from the non-tolerant population (N0) (Figure 4). The qRT–PCR results further confirmed that this significant differential expression (*p* < 0.05) involves the regulation of transcript levels (Figure 5). CWIN has attracted much attention for its effect on plant yield and environmental tolerance. Likewise, exogenous SA application can also directly or indirectly influence yield and increase stress tolerance. Khan et al. showed that SA improves photosynthesis and growth under salt stress in mungbean [54], and SA also improved growth and increased Cd tolerance in rice [55]. Similar results have been reported in flax, bluegrass, and radish [56]. However, antisense repression of *CWIN* impaired carrot root development [57]. In *Zea mays*, SA levels were increased 10-fold in the cell wall invertase (INCW2) mutant *mn1* relative to the WT [58]. These findings suggest a degree of crosstalk between SA and *CWIN* expression. Our results provide further confirmation that SA regulates *CWIN* promoter activity, potentially leading to differences in CWIN activity and gene expression between the different populations. This finding provides new insight into the regulation of *CWIN* transcript levels in Cu-tolerant and non-tolerant *E. haichowensis* populations.

### 3.4. A 270-bp Promoter Fragment Is Required for the SA Response

The regulation of gene transcription involves the binding of TFs to specific *cis* elements. SA significantly induced the activity of the *CWIN* promoter from the Cu-tolerant population but not from the non-tolerant population. Therefore, we speculated that a *cis* element or enhancer might be present in the *CWIN* promoter from the tolerant population. Based on *cis* elements that differed between the two promoters, we made four different deletion constructs (C1–C4) from the full-length Cu-tolerant promoter (Figure 1B). SA did not induce the activity of any of the promoter deletions (Figure 6). Therefore, we concluded that the region deleted in C1 but present in C0 was the key fragment influenced by SA. This 270-bp region spanned −1729 to −1460 bp upstream of the start codon. When this region was compared between promoters from the two populations, they were found to share common TCA, LTR, STRE, and CuRE elements (Figure 1B). However, there were 10 different sites between the two promoters, including eight SNPs and two inserted mutations in the promoter from the Cu-tolerant population. These two inserted mutations contain putative W-box and CAAT-box *cis*-elements (Figure 1A). Interestingly, the CAAT-box insertion mutation was located 16 bp upstream of an SA *cis* element (TCA element). A number of studies have provided evidence that the CAAT-box (consensus sequence CCAAT) is usually associated with enhanced promoter activity in plants [59,60], viruses [61], insects [62], and mammals [63]. Enhancer–promoter regulation is a fundamental mechanism underlying differential transcriptional regulation [64], especially when the enhancer and promoter are in close enough proximity to interact directly, without the need for facilitating mechanisms [65]. In addition, the binding of WRKY proteins to the W-box is also thought to be related to regulation by SA. Mohr et al. reported that Arabidopsis *RPP8* is induced by SA and that this process is regulated by the W-box [66]. Expression of the *PR1* gene was induced by SA and regulated synergistically by WRKY50 and TGA TFs in Arabidopsis [67]. These findings imply that the regulation of SA response is complex. In future work, we will further shorten the promoter, or mutate the CAAT-box and W-box, in order to analyze the promoter in greater detail.

In conclusion, we have isolated the *CWIN* promoter from Cu-tolerant and non-tolerant populations of *E. haichowensis*. Two full-length promoter–reporter vectors and four 5′-deletion promoter–reporter vectors were constructed and transformed into Arabidopsis. Copper slightly increased the activity of the *CWIN* promoter from tolerant populations. Glucose and fructose significantly induced (*p* < 0.05) the activity of *CWIN* promoters from both populations, perhaps because of the presence of W-box and SP8 sugar-induced response elements. ABA slightly inhibited promoter activity, whereas SA, 6-BA, GA, and ACC slightly stimulated it. Only SA induced significantly higher (*p* < 0.05) activity in the Cu-tolerant promoter than in the non-tolerant and 35S promoters, and SA concentrations of 100–1000 μM induced significantly (*p* < 0.05) higher C0 promoter activity. Likewise, qRT–PCR analyses showed that *GUS* gene expression was significantly higher (*p* < 0.05) in C0 lines than in N0 and 35S lines at 6 h and 12 h. A 270-bp promoter sequence appeared to be required for SA induction of C0 promoter activity. There were eight SNP sites and two InDels in this region., and CAAT-box (CCAAT) and W-box (TTGAC) *cis* element inserted mutations were found in the Cu-tolerant promoter only. These differences in promoter sequences may have driven the previously observed differences in *CWIN* expression between the two populations. Nonetheless, we cannot exclude the possibility that *CWIN* is regulated post-transcriptionally or post-translationally. In summary, our study suggests that the activity of the *CWIN* promoter is significantly induced (*p* < 0.05) by glucose, fructose, and SA in the Cu-tolerant population. The CAAT-box and W-box insertion mutation in the Cu-tolerant promoter sequence may have caused changes in SA induction, thereby increasing the activity of CWIN in Cu-tolerant populations. This study provides evidence for a regulatory role of SA in *CWIN* gene expression at the transcriptional level and reveals new insights into differences in *CWIN* expression between *E. haichowensis* populations.

## 4. Materials and Methods

### 4.1. Bioinformatics Analysis of Promoters

In our previous studies, we cloned the *CWIN* genes from Cu-tolerant (*EhCcwINV*, Genbank No. JX500754) and non-tolerant (*EhNcwINV*, Genbank No. JX500753) populations of *E. haichowensis* [17,18]. Their promoter regions were isolated from genomic DNA by the hiTAIL-PCR technique [68]. The full-length Cu-tolerant promoter (*EhCcwINVP*) and non-tolerant promoter (*EhNcwINVP*) were determined to be 1732 bp and 1727 bp in length and were given GenBank accession numbers KC985183 (*EhCcwINVP*) and KC985182 (*EhNcwINVP*), respectively. The *cis* elements of the promoters were analyzed using the online PlantCARE Available online: (http://bioinformatics.psb.ugent.be/webtools/plantcare/html/, accessed on 18/5/2016) [69] and PLACE databases (Available online: https://www.dna.affrc.go.jp/PLACE/?action=newplace, accessed on 12/6/2020) [70].

### 4.2. Construction of Plant Expression Vectors and Plant Transformation

For *β*-glucuronidase (GUS) reporter gene analysis, full-length promoter and 5′-deletion promoter constructs of different sizes were obtained by PCR amplification [20]. The construction of plant expression vectors was performed as described by Zhao et al. [71]. The fusion constructs were designated as follows: full-length promoters *EhCcwINVP:GUS* (C0, −1729 to ATG) and *EhNcwINVP::GUS* (N0, −1724 to ATG). Based on the different structural characteristics and distribution of *cis* elements between C0 and N0, the 5′-deleted promoters fusion constructs, C1 (−1460 to ATG), C2 (−729 to ATG), C3 (−612 to ATG), and C4 (−209 to ATG) were constructed from C0, respectively (Figure 1B). The primers for the promoter clone are shown in Figure 1A. These recombinant binary vector plasmids were mobilized into *Agrobacterium tumefaciens* after confirmation by restriction analysis, and glycerol stocks were stored at −80 °C until further use. Sequencing was performed by the BGI Company (Wuhan China). For subsequent reporter gene expression studies, the *CaMV35S* promoter was used as the positive control (35S), and Columbia wild-type (WT) *Arabidopsis thaliana* was used as the negative control. The recombinant plasmids in *A. tumefaciens* were transformed into Arabidopsis using the floral dip method [72], and at least three transgenic lines were obtained for each construct in T_3_ transformants.

### 4.3. Plant Materials, Stress Treatments, and Sampling

The transgenic Arabidopsis seedlings were vernalized after transplant for 3 days in 1/2 Murashige and Skoog (MS) medium under a regime of light (122 μM m^−2^ s^−1^)/dark 16/8 h and 25/22 °C in a growth chamber. Fourteen-day-old seedlings were used for the following treatments. For the Cu treatment, seedlings were transferred to solid 1/2 MS medium that contained 0.3 μM (control), 25 μM, 50 μM, 100 μM, or 200 μM Cu^2+^ (CuCl_2_·2H_2_O). For sugar treatments, the seedlings were transferred to new solid MS medium without sugar, cultured for 24 h in the dark, and then submerged in aqueous 1/2 MS medium supplemented with 1/2 MS liquid medium (control) or 1/2 MS liquid medium containing sucrose (Suc, 200 mM), glucose (Glc, 200 mM), fructose (Fru, 200 mM), fructose + glucose (F + G, 100 mM, 1:1), or mannitol (Man 200 mM, osmotic control) [73]. For hormone treatments, seedlings were exposed to ABA, MeJA, SA, 6-BA, GA, IAA, or ACC. These phytohormones used a small amount of 80% ethanol to aid solubility except cytokinin (6-BA). For 6-BA, we first used a small amount of 0.1 M hydrochloric acid. Then, we used autoclaved ultrapure water to make up the stock solutions [74,75,76]. All stock solutions were freshly prepared at room temperature, and dilutions were prepared using distilled water. The seedlings were transferred to filter paper soaked with 20 μM ABA, 20 μM 6-BA, 20 μM IAA, 20 μM GA, 100 μM ACC, 100 μM MeJA, 100 μM SA, or sterile water (control, CK). In addition, a series of SA concentrations (0–1000 μM) was used to further evaluate the promoter activity of the full-length constructs. *A. thaliana* seedlings were treated for 24 h. After treatment, seedlings were harvested for quantitative analysis, including GUS activity, total protein, and quantitative real-time PCR (qRT-PCR). Each assay was performed on at least three independent lines and repeated three times. Individual samples of 8–10 seedlings were harvested, quickly frozen in liquid nitrogen, and stored at −80 °C.

### 4.4. Histochemical Staining and Fluorometric Quantification of GUS Activity

GUS activity was assessed by histochemical staining and fluorometric measurement of transgenic Arabidopsis as described by Jefferson [77], with some modifications. Then, various tissues were observed under a stereoscope and a light microscope. Blue staining was interpreted as a positive sign of GUS expression. For fluorometric measurement, plant material was homogenized in 50 mM sodium phosphate (pH 7.0), 10 mM Na_2_EDTA (pH 8.0), 0.1% (*v/v*) Triton X-100, 0.1% (*w/v*) SDS, 10 mM β-mercaptoethanol, and 20% (*v/v*) methanol; then, it was centrifuged for 20 min at 12,000 rpm. The supernatant was used for protein quantification and the fluorometric assay. After reaction with 4-methylumbelliferyl glucuronide (4-MUG), fluorescence was measured using a Cytation 3 Cell Imaging Multi-Mode Reader (BioTek, Winooski, USA) set at excitation and emission wavelengths of 365 and 455 nm. Protein concentration was determined by the Bradford method using bovine serum albumin as a standard [78], and the OD value was measured at 595 nm. The amount of 4-methylumbelliferone (4-MU) released by enzymatic activity was determined from a standard curve, and GUS enzyme activity was expressed as pmol 4-MU/min/mg protein.

### 4.5. RNA Extraction and qRT–PCR Analysis of GUS Gene Expression

Total RNA was extracted from tissue samples using the TRNzol Universal reagent (Tiangen, Beijing, China) according to the manufacturer’s instructions. The quality of the total RNA was checked spectrophotometrically, and its molecular integrity was assessed by gel electrophoresis. The cDNA for qRT–PCR was synthesized from RNA samples using the PrimeScript RT reagent Kit with gDNA Eraser (TaKaRa, Dalian, China), and qRT–PCR was performed using SYBR Premix Ex TaqII (TaKaRa) on the Step One Real-Time PCR System (Applied Biosystems) according to the standard protocol. The qRT–PCR conditions were as follows: 94 °C for 30 s, 40 cycles of 95 °C for 5 s and 60 °C for 30 s, melting curve from 60–95 °C. qRT–PCR detection of the *GUS* gene was performed with primers GUS-Forward: AACGGGGAAACTCAGCAAGC and GUS-Reverse: TCCGGTTCGTTGGCAATACTC. Arabidopsis *Actin 2* (At3g18780) was used as the internal reference gene (primers Act2-Forward: CTTGCACCAAGCAGCATGAA and Act2-Reverse: CCGATCCAGACACTGTACTTCCTT). The experiment was performed with three biological and three technical replicates.

### 4.6. Statistical Analysis

The data are presented as mean ± SD of three independent experiments. Statistical analysis was performed with one-way ANOVA using the IBM SPSS Statistics 20. Different letters indicate a significant difference (*p* < 0.05) according to Duncan’s test (*n* = 3).

## Figures and Tables

**Figure 1 ijms-22-05299-f001:**
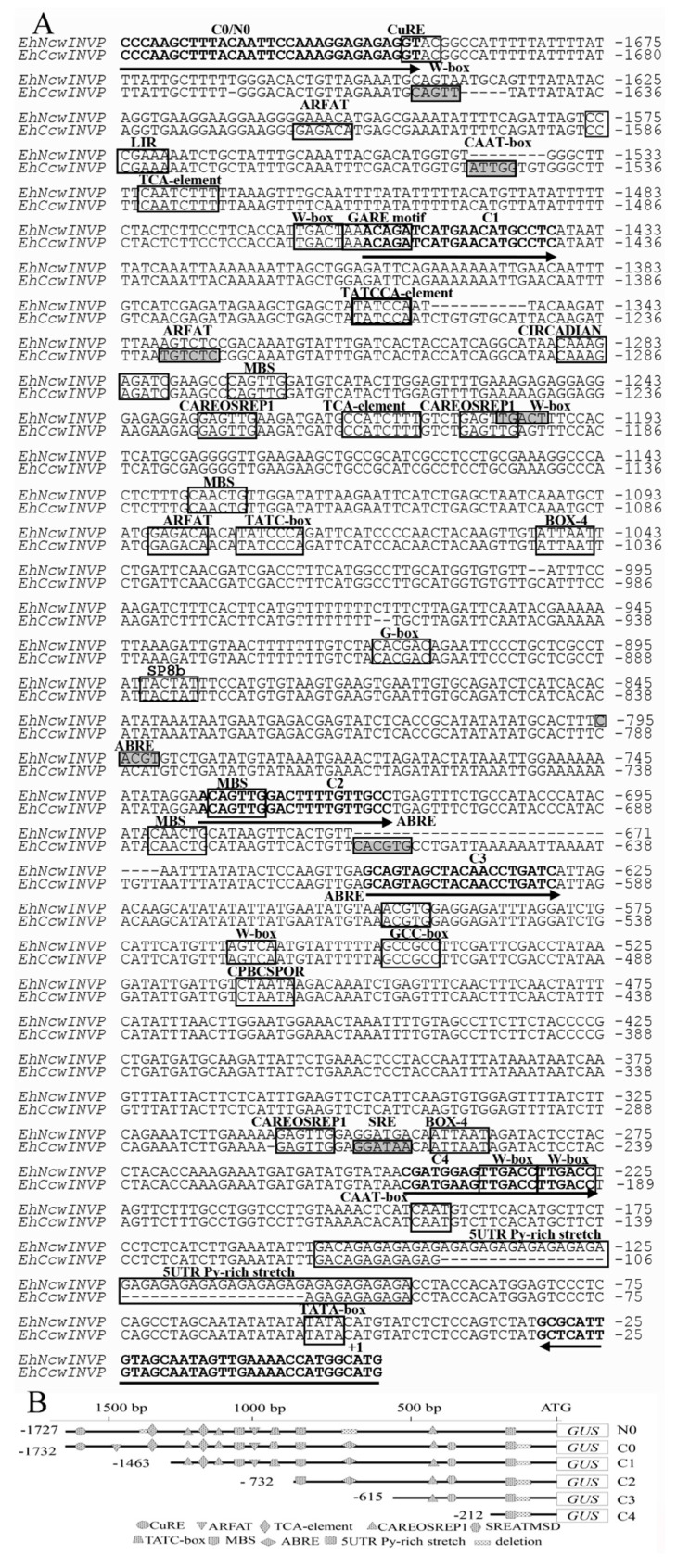
The putative *cis* elements of *EhCcwINVP* and *EhNcwINVP* (**A**) and schematic diagrams of the full-length and 5′-deletion *CWIN* promoters that drove a GUS-reporter gene (**B**). In (**A**), the *cis* elements are framed in boxes and labeled above. The *cis* elements with a gray background are unique to one promoter. The forward primers and the common reverse primer for cloning the full-length promoter and the four 5′-deletion promoters are in bold and are indicated by arrows. The transcription start site (ATG) is designated as +1. In (**B**), the shaded shapes represent promoter *cis* elements. N0 and C0 are GUS fusion constructs with the full-length promoters from non-tolerant and Cu-tolerant populations, respectively. C1, C2, C3, and C4 are 5′-deletion GUS fusion constructs of the promoter from the Cu-tolerant population.

**Figure 2 ijms-22-05299-f002:**
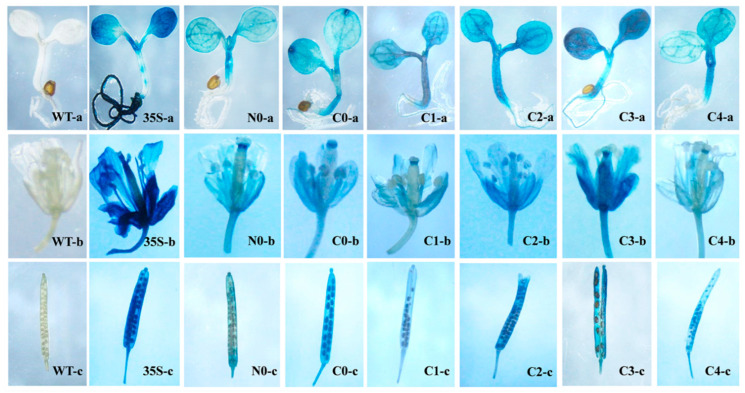
Histochemical staining of GUS activity. WT, wild-type Arabidopsis served as the negative control. 35S, lines containing the CaMV35S promoter served as positive controls. N0 and C0 were full-length promoter transformants of *EhNcwINVP* and *EhCcwINVP*, respectively. C1, C2, C3, and C4 were transformants carrying 5′ deletions of *EhCcwINVP* (−1460, −729, −612, and −209 bp upstream of ATG, respectively). a: five-day-old seedlings, b: flowers, c: silique pods.

**Figure 3 ijms-22-05299-f003:**
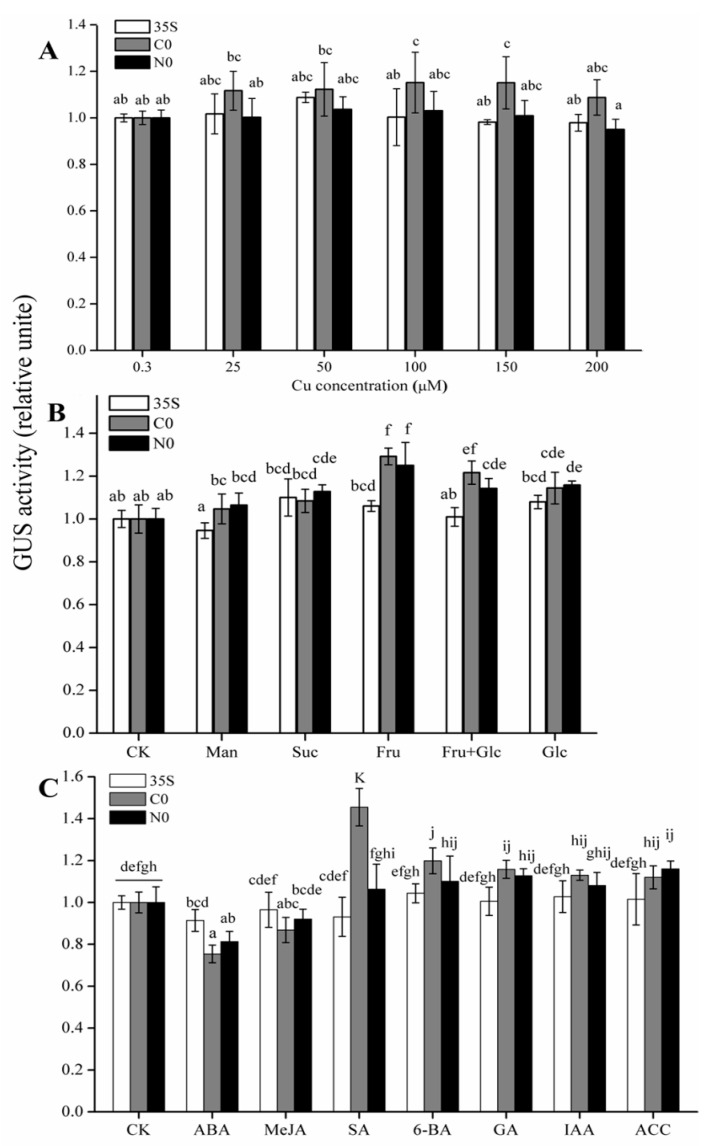
Relative GUS activity of full-length promoter constructs under different treatments. (**A**). Relative GUS activity in 35S, C0, and N0 lines following treatment with 0.3 μM (control), 25 μM, 50 μM, 100 μM, 150 μM, or 200 μM Cu^2+^ (CuCl_2_·2H_2_O). (**B**). Relative GUS activity in 35S, C0, and N0 lines following treatment with ½ MS liquid medium (CK) or ½ MS liquid medium containing sucrose (Suc, 200 mM), glucose (Glc, 200 mM), fructose (Fru, 200 mM), fructose+glucose (F+G, 100 mM, 1:1), and mannitol (Man, 200 mM, osmotic control). (**C**). Relative GUS activity in 35S, C0, and N0 lines following treatment with 20 μM abscisic acid (ABA), 20 μM cytokinin (6-BA), 20 μM auxin (IAA), 20 μM gibberellin (GA), 100 μM ethylene precursor (1-aminocyclopropane-1-carboxylic acid, ACC), 100 μM methyl jasmonate (MeJA), 100 μM salicylic acid (SA), or sterile water (CK). Relative GUS activity is the GUS activity of each treated transgenic line relative to its control, and bars represent the mean ± SD. Different letters indicate significant differences (*p* < 0.05) according to one-way ANOVA followed by Duncan’s test (*n* = 3).

**Figure 4 ijms-22-05299-f004:**
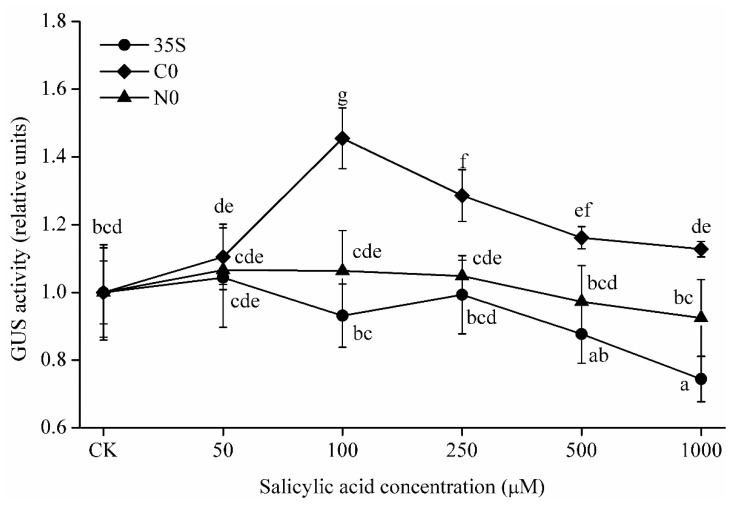
Relative GUS activity in 35S, C0, and N0 lines at a series of exogenous SA concentrations (0, 50, 100, 250, 500, and 1000 μM). Relative GUS activity was calculated as the activity of the treated transgenic line divided by the activity of its untreated control. Bars represent the mean ± SD, and different letters indicate significant differences (*p* < 0.05) based on Duncan’s test (*n* = 3).

**Figure 5 ijms-22-05299-f005:**
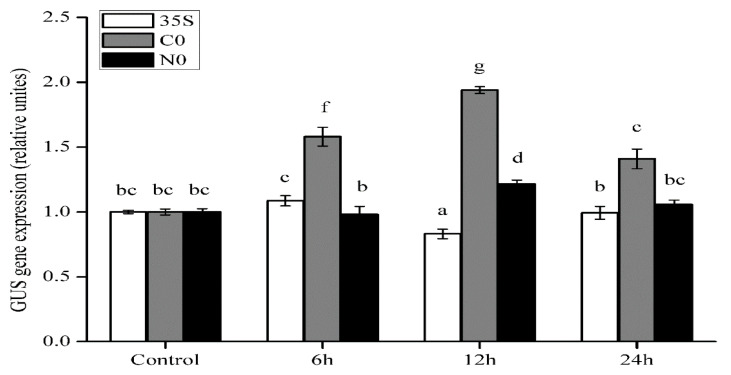
qRT-PCR analysis of GUS gene expression driven by full-length promoter constructs under salicylic acid treatment. Fourteen-day-old Arabidopsis seedlings were exposed to 100 μM SA for 6, 12, or 24 h. Seedlings were used for RNA extraction and qRT–PCR analysis. Data represent the mean ± SD of three independent transgenic lines, each measured with three technical replicates. Different letters indicate significant differences (*p* < 0.05) according to one-way ANOVA followed by Duncan’s test (*n* = 3).

**Figure 6 ijms-22-05299-f006:**
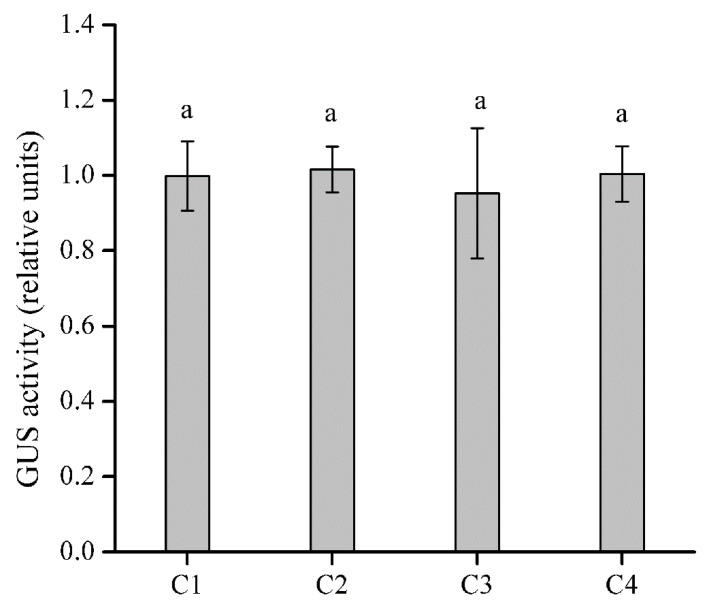
Relative GUS activity of transgenic Arabidopsis harboring 5′-deletion promoter constructs under 100 μM SA treatment. The 5′-deletion promoter constructs were C1 (−1460 to ATG), C2 (−729 to ATG), C3 (−612 to ATG), and C4 (−209 to ATG). Relative GUS activity was calculated as the activity of the treated transgenic line divided by the activity of its untreated control. Bars represent the mean ± SD, and different letters indicate significant differences (*p* < 0.05) based on Duncan’s test (*n* = 3).

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
