# Peer review of "Comparison and Characterization of a Cell Wall Invertase Promoter from Cu-Tolerant and Non-Tolerant Populations of *Elsholtzia haichowensis"

_ijms, 2021, doi:10.3390/ijms22105299_

Round 1

Reviewer 1 Report

Dear authors, The manuscript presents very interesting data on the regulation and control of the promoters of invertase genes, whose role in the development and tolerance of environmental stresses is quite emphasized. However, some important points should be addressed:
1.
The amount of sugars used (200 mM) appears to be too high for plant growth. Couldn't this treatment have caused an osmotic imbalance?
2.
Please indicate how the hormones were prepared. The solubilization of hormones is extremely important to make them available to plants. There would be a possibility that they did not have the expected effect because of how they were prepared and applied
3.
The discussion section should be better prepared. For a moment it seems that I've been reading materials and methods, the summary of the work. The conclusions are not adequate.
4.
I suggest making comparisons with other studies on plants and not with human populations (in the discussion).
5.
Why did the different sugars tested have no effect on the regulation of invertase promoters? A comparison of the sequence of promoters of E. haichowensis populations with those that are induced by sugars could perhaps reveal differences. The authors should discuss this point.
6.
In item 3.1 of the discussion: however, our data do not support heavy metal or sugar regulation ... the results do not support; however, what can be the explanation? Can the comparison of promoter sequences of different species show differences in regulatory elements that explain why promoters of the studied species are not induced by heavy metals?
7.
The authors point out in the discussion: ... The activity of the CWIN promoter from the Cu tolerant population did not differ from that of the non-tolerant and 35S promoter ... Are the differences indicated in relation to function or structure?
8.
Do the authors have any notion of why the promoters for the cell wall invertase gene would have diverged in the Cu tolerant and non-tolerant populations?

Author Response

Reviewers #1

      1.The amount of sugars used (200 mM) appears to be too high for plant  growth. Couldn't this treatment have caused an osmotic imbalance?

Response: thanks reviewer for pointing this out. The sugar concentration of 200mm is a relatively high concentration for the treatment of plants in liquid medium. Murashige and Skoog medium (MS) was a commonly used liquid medium for tissue or seeding and its sucrose concentration is 3% (87.6mM)(Murashige and Skoog 1962). In our research, to observe the influence of sugar on promoter activity, considering the influence of osmotic pressure, we choose 1/2MS (no sugar) medium, and hope to treat with higher sugar concentration than normal MS medium, so that to observe the GUS expression of the promoter induced by sugar. Therefore, 200mM sucrose was selected, at the same time, our experiment also has an osmotic control. Mannitol is often used as an osmotic control in liquid medium (Zhong et al., 2015). Based on our data (Figure 3, B), the mannitol (Man) did not cause significant changes in GUS enzyme activity. Therefore, we believe that 200mm of sugar may not cause an osmotic imbalance. Some similar experiments have been reported that 6% (175mM) to 200mM sugar were used to treat plants in MS liquid medium(Cho and Yoo 2011; Wu, Lv, and Liu 2007; Zhong et al. 2015), even 300mM(Tsukaya et al. 1991), which results showed that the plant also can normal growth.

  1. Please indicate how the hormones were prepared. The solubilization of hormones is extremely important to make them available to plants. There would be a possibility that they did not have the expected effect because of how they were prepared and applied

Response: we would like to thank the reviewer very much for the kind evaluation and suggestions for our manuscript.  For abscisic acid (ABA), auxin (IAA), gibberellin (GA), ethylene precursor (1-aminocyclopropane-1-carboxylic acid, ACC), methyl jasmonate (MeJA), salicylic acid (SA), We first use a small amount of 80% ethanol to aid solution; For cytokinin (6-BA), We first use a small amount 0.1M·L-1 hydrochloric acid solubilization; and then use autoclaved ultrapure water to make the mother liquor. These methods have been applied in similar previous literature(Liu, WU, and Hao 2001; Spoel et al. 2003; Van der Does et al. 2013). The mother liquor was diluted 1000 times before use. We have also paid attention to some properties of plant hormones, such as Gibberellic is poorly soluble in water and well soluble in organic solvents(Poprotska et al. 2019), and has thermal instability(Gana 2011). IAA has a photodegradation effect(Gana 2011). Therefore, what we use in processing plants is freshly prepared and used, and experiments are carried out at room temperature. We have added related description in the revised manuscript.

  1. The discussion section should be better prepared. For a moment it seems that I've been reading materials and methods, the summary of the work. The conclusions are not adequate.

Response: these suggestions of reviewers in this part are very important for us to improve our manuscript. We have rewritten the manuscript substantially to address the Reviewers’ concerns and incorporate the new findings. In the discussion subchapter 3.1, we compared the relationship between copper response and copper cis-elements in the previous plant researches, and discussed the possible mechanism of copper response in this study. In the discussion subchapter 3.2, we briefly described the sugar response cis-elements of plants and discussed the cis-acting elements in response to glucose and fructose in this study. We have also rewritten the conclusion chapter and added new findings that copper and sugar regulate the activity of the CWIN promoter of Elsholtzia haichowensis in copper-tolerant populations.

  1. I suggest making comparisons with other studies on plants and not with human populations (in the discussion).

Response: we agree with the reviewer’s suggestions, we have updated the manuscript and focused on the discussion about plants.

  1. Why did the different sugars tested have no effect on the regulation of invertase promoters? A comparison of the sequence of promoters of E. haichowensis populations with those that are induced by sugars could perhaps reveal differences. The authors should discuss this point.

Response: we thank the reviewers for their constructive and insightful suggestions and comments. We did not clearly express our views in the previous manuscript. In the revised manuscript, we re-describe the results of sugar treatment. Glucose and fructose can significantly induce the promoter activity of the two populations, but the promoter activity of both populations are induced by glucose and fructose, so there is no significant difference in glucose and fructose between the two populations.

  1. In item 3.1 of the discussion: however, our data do not support heavy metal or sugar regulation ... the results do not support; however, what can be the explanation? Can the comparison of promoter sequences of different species show differences in regulatory elements that explain why promoters of the studied species are not induced by heavy metals?

Response: thanks to the reviewer for correcting our expression mistake. We have revised this expression in the results and discussion section. At the transcription level, the activity of the promoter requires the joint action of cis-elements and transcription factors (Shrestha, Khan, and Dey 2018). In the discussion subchapter 3.1, Copper do not induce differential expression of CWIN promoters from the Cu-tolerant and non-tolerant populations, we tried to discuss that there were no significant difference in promoter activity between the two populations, including single copies of copper response elements (CuRE), enhancers and promoter methylation modifications. In the discussion subchapter 3.2, Glucose and fructose significantly induced the activity of the CWIN promoter both in Cu-tolerant and non-tolerant populations, we discussed possible cis-elements induced by sugar both the two populations.

  1. The authors point out in the discussion: ... The activity of the CWIN promoter from the Cu tolerant population did not differ from that of the non-tolerant and 35S promoter ... Are the differences indicated in relation to function or structure?

Response: The difference in promoter activity implies a structural difference, but it is not absolute. It is necessary to further determine whether caused by structural differences through 5'-end or 3'-end deletion analysis. However, the regulation of plant gene expression involves multiple regulatory layers, including transcriptional, post-transcriptional, post-translational, and metabolic regulation (Haak et al. 2017). We had added these contents in the discussion section.

  1. Do the authors have any notion of why the promoters for the cell wall invertase gene would have diverged in the Cu tolerant and non-tolerant populations?

Response: Local adaptation of plant populations can produce heritable variation (Nagy 1997). We speculate that the Cu-tolerant population of Elsholtzia haichowensis has long been adapted to the high-copper environment, its root need more energy for the roots to maintain the stress resistance mechanism. In root, sucrose was decomposed into glucose and fructose under the action of cell wall invertase to supply reservoir organs for growth, development and response to environmental stimulus. The severe environments of ancient copper mines are relatively nutrient deficiencies for plants (Cai et al. 2014; Li et al. 2007). To deal with the nutrient deficiencies environments, the Cu tolerant populations of Elsholtzia haichowensis needs to evolve into a more efficient use of nutrients. Sucrose (Suc) is the major product of photosynthesis in the source tissues of higher plants; it is transported to sink tissues to sustain cell metabolism, growth, and environmental stress tolerance (Sturm and Tang 1999; Rolland, Baena-Gonzalez, and Sheen 2006). Cell wall invertase plays a central role in the decomposition and utilization of sucrose in plants (Proels and Huckelhoven 2014). The root cell wall invertase (CWIN) activity and CWIN gene expression were significantly higher (P<0.05) in the Cu-tolerant population than in the non-tolerant population under Cu stress (Cai et al. 2014). However, the CWIN genes from the two populations had extremely similar deduced amino acid sequences and predicted 3D structures (data not shown). Therefore, we believe that different sucrose utilization may involve promoter regulation at the transcriptional level.

Reviewer 2 Report

Manuscript ˝Comparison and characterization of a cell wall invertase promoter from Cu-tolerant and non-tolerant populations of Elsholtzia haichowensis˝ needs extensive modifications of the organization and presentation of the results given in the article.

The organization of paper in the way that Results are given first, followed by discussion, with Materials and Methods at the end of the paper, demands somewhat different style of writing than the one that authors offered here to be comprehensible. Authors wrote the paper in the way that would be acceptable for the form of paper in which the Materials and Methods are given first, followed by results and with discussion at the end. However, in this form, short introduction to every experiment shown in the Results chapter should be given, in which the reasoning behind the experiment as well as short description of methodology, strains etc. used in the experiment. Without that it is hard to understand and follow what has been done and with which reason.  

There are many abbreviations that are not explained when first time used in text. They are not self-explanatory, so reader must search for explanations through the figures and the text further on in the paper.  All abbreviations should be shortly explained when first time mentioned. Also, concepts and terms should be briefly explained when they are first mentioned.

For example:

Abstract – What is SA?

Results subchapter 2.1  -  what are ˝EhNcwINVP and EhCcwINVP˝?

Results subchapter 2.2 -  what are ˝35S, N0, C0, C1, C2, C3, and C4˝ ?

Results subchapter 2.3  -  explain the meaning of the statement that plants  ˝received sugar treatments (Suc, Fru, Fru+Glc, and Glc)˝?

                                        - what are ˝mannitol group and the CK group˝?

                                        - what are ˝ABA, MeJA, SA, 6-BA, GA, IAA, or ACC˝?

Furthermore, Figure 1 and Fig 7 should be combined in one figure.

There are some sentences in the text starting with ˝And…˝ - syntax should be corrected in these cases.

The sentence in chapter Discussion, ˝Promoters are crucial roles in the regulation of gene expression˝ is meaningless. ˝Promoters have crucial roles˝, or something alike would have sense.

In the first sentence of 4.1. subchapter (Materials and Methods), authors declared that they have cloned ˝the CWIN genes from Cu-tolerant (EhCcwINV, Genbank No. JX500754) and non-tolerant (EhNcwINV, Genbank No. JX500753)….˝. In the third sentence they stated that ˝The full-length Cu-tolerant promoter (EhNcwINVP) and non-tolerant promoter (EhCcwINVP) were determined….˝. Which gene belongs to tolerant and which one to non-tolerant strain at the end?

Furthermore, authors should explain what was the reason to use CaMV35S promoter as positive, and wt A. thaliana as negative control in the experiment. (For subsequent reporter gene expression studies, the CaMV35S promoter was used as the positive control (35S), and Columbia wild-type (WT) Arabidopsis thaliana was used as the negative control.)

Manuscript might be acceptable for publication after reorganization and clarification of the text.

Author Response

Reviewers #2

1.The organization of paper in the way that Results are given first, followed by discussion, with Materials and Methods at the end of the paper, demands somewhat different style of writing than the one that authors offered here to be comprehensible. Authors wrote the paper in the way that would be acceptable for the form of paper in which the Materials and Methods are given first, followed by results and with discussion at the end. However, in this form, short introduction to every experiment shown in the Results chapter should be given, in which the reasoning behind the experiment as well as short description of methodology, strains etc. used in the experiment. Without that it is hard to understand and follow what has been done and with which reason. 

Response: we are very grateful to reviewers for your very practical suggestions on manuscript writing, which were very important for us to improve our manuscript. We have meticulous organized the manuscript in accordance with the reviewers’ suggestions and the journal editor’s requirements. Following this order Introduction, Results, Discussion, Materials and Methods give at the end. At the results chapter, we first give a reason for the experiment, and then a brief introduction for materials and conditions was given. These improvements have been marked as green.

  1. There are many abbreviations that are not explained when first time used in text. They are not self-explanatory, so reader must search for explanations through the figures and the text further on in the paper.  All abbreviations should be shortly explained when first time mentioned. Also, concepts and terms should be briefly explained when they are first mentioned.

For example:

Abstract – What is SA?

Results subchapter 2.1  -  what are ˝EhNcwINVP and EhCcwINVP˝?

Results subchapter 2.2 -  what are ˝35S, N0, C0, C1, C2, C3, and C4˝ ?

Results subchapter 2.3  -  explain the meaning of the statement that plants  ˝received sugar treatments (Suc, Fru, Fru+Glc, and Glc)˝?

                                        - what are ˝mannitol group and the CK group˝?

                                        - what are ˝ABA, MeJA, SA, 6-BA, GA, IAA, or ACC˝?

Response: We are very sorry that we did not explain when these abbreviations first appeared. Following the reviewer’s constructive suggestions, we have added these contents in the abstract chapter and results chapter, which will help readers to better understand this article. We have corrected all abbreviation, when the first time, the full word used with the abbreviation between brackets, afterwards only the abbreviation should be used

Abstract – What is SA?

SA was a abbreviations of salicylic acid. We have modified the description in Abstract chapter as follows:“only salicylic acid (SA)induced significantly higher (P<0.05) activity of the Cu-tolerant CWIN promoter relative to the non-tolerant. Analysis of 5'-deletion constructs revealed that a 270-bp promoter fragment was required for SA induction of the promoter from the Cu-tolerant population.”

Results subchapter 2.2  - 

- what are ˝EhNcwINVP and EhCcwINVP˝?

We have modified a description in the Results subchapter 2.1 as follows:“To analyze differences in CWIN expression between Cu-tolerant and non-tolerant populations under Cu stress, we used the Clustal W multiple sequence aligner to compare the full-length sequences of CWIN promoter of Cu-tolerant population (EhCcwINVP) and the full-length sequences of CWIN promoter of non-tolerant population (EhNcwINVP) of E. haichowensis.”

Results subchapter 2.2 - 

-what are ˝35S, N0, C0, C1, C2, C3, and C4˝ ?

We have added a description within the Results subchapter 2.2 as follows:“In order to further identify possible response fragments of promoter with different activity between C0 and N0, four 5'-deleted promoters fusion were constructed from C0, including C1 (−1460 to ATG), C2 (−729 to ATG), C3 (−612 to ATG), and C4 (−209 to ATG) (Figure 1, A, B). The constitutive expression CaMV35S promoter (CaMV35S::GUS) was used as the positive control (35S), and Arabidopsis wild-type (no GUS gene) was used as the negative control (WT).”

Results subchapter 2.3  - 

- explain the meaning of the statement that plants  ˝received sugar treatments (Suc, Fru, Fru+Glc, and Glc)˝?

We have given a brief introduction the sugar treatments processing in the results subchapter 2.3 as follows: “For the sugar treatment, sucrose (Suc, 200 mM), glucose (Glc, 200 mM), fructose (Fru, 200 mM), fructose+glucose (Fru+Glc, 100 mM, 1:1), mannitol (Man 200 mM, osmotic control) and 1/2 MS liquid medium (no sugar, CK) were used treating the transgenic Arabidopsis”.

Results subchapter 2.3  - 

                                        - what are ˝mannitol group and the CK group˝?

Sugar treatment of transgenic Arabidopsis in 1/2 MS liquid medium may cause osmotic effects. Therefore, in the sugar treatment experiment, we have set up two controls including osmotic control and sample blank control to illustrate the effect of osmotic pressure on promoter activity. We have added a description within the Results subchapter 2.3 as follows: “In addition, there were no significant differences between the mannitol group (osmotic control) and the CK group (blank control) for C0 or N0, implying that the CWIN promoter was not regulated by osmotic pressure”.

Results subchapter 2.3  - 

                                        - what are ˝ABA, MeJA, SA, 6-BA, GA, IAA, or ACC˝?

We have added a description within the Results subchapter 2.3 as follows:“In the hormone treatments, exogeneous abscisic acid (ABA), auxin (IAA), gibberellin (GA), ethylene precursor (1-aminocyclopropane-1-carboxylic acid, ACC), methyl jasmonate (MeJA), salicylic acid (SA) and sterile water (control, CK) was applied to the transgenic Arabidopsis seedlings”.

  1. Figure 1 and Fig 7 should be combined in one figure.

Response: we agree with the reviewers.

  1. There are some sentences in the text starting with ˝And…˝ - syntax should be corrected in these cases.

Response: We have scrutinized the manuscript, and made according revisions including some typos, grammatical errors and long sentences, etc., and the manuscript was sent to a language editing agency for two rounds of editing by native English experts.

  1. The sentence in chapter Discussion, ˝Promoters are crucial roles in the regulation of gene expression˝ is meaningless. ˝Promoters have crucial roles˝, or something alike would have sense.

Response: We thank and agree with the reviewers’ very appropriate suggestions.

  1. In the first sentence of 4.1. subchapter (Materials and Methods), authors declared that they have cloned ˝the CWIN genes from Cu-tolerant (EhCcwINV, Genbank No. JX500754) and non-tolerant (EhNcwINV, Genbank No. JX500753)….˝. In the third sentence they stated that ˝The full-length Cu-tolerant promoter (EhNcwINVP) and non-tolerant promoter (EhCcwINVP) were determined….˝. Which gene belongs to tolerant and which one to non-tolerant strain at the end?

Response: We are sorry that the gene description for belongs to Cu-tolerant or non-tolerant is not clear. We have added a description in the third sentence of 4.1. subchapter as follows:“The full-length Cu-tolerant population promoter (EhNcwINVP) and non-tolerant population promoter (EhCcwINVP) were determined to be 1727 bp and 1732 bp in length and were given GenBank accession numbers KC985183 (EhNcwINVP) and KC985182 (EhCcwINVP), respectively.”

  1. authors should explain what was the reason to use CaMV35S promoter as positive, and wt A. thaliana as negative control in the experiment. (For subsequent reporter gene expression studies, the CaMV35S promoter was used as the positive control (35S), and Columbia wild-type (WT) Arabidopsis thaliana was used as the negative control.)

Response: We agree with the reviewer's suggestion, we have added a description in 4.2 as follows: “For subsequent reporter gene expression studies, the CaMV35S promoter was used as the positive control (35S) because it is a strong and constitutive expression promoter, The Columbia wild-type (WT) Arabidopsis thaliana was used as the negative control because there is no GUS gene in its body.”
